# Zn–Mn-Doped Mesoporous Bioactive Glass Nanoparticle-Loaded Zein Coatings for Bioactive and Antibacterial Orthopedic Implants

**DOI:** 10.3390/jfb13030097

**Published:** 2022-07-16

**Authors:** Syeda Ammara Batool, Khalil Ahmad, Muhammad Irfan, Muhammad Atiq Ur Rehman

**Affiliations:** 1Department of Materials Science and Engineering, Institute of Space Technology Islamabad, Islamabad 44000, Pakistan; syedaammara001@gmail.com (S.A.B.); k.ahmad644@gmail.com (K.A.); 2School of Chemical and Materials Engineering (SCME), National University of Sciences and Technology (NUST) H-12, Islamabad 44000, Pakistan; muhammad.irfan@scme.nust.edu.pk

**Keywords:** electrophoretic deposition, mesoporous bioactive glass nanoparticles, adhesion, antibacterial, bioactive

## Abstract

In recent years, natural polymers have replaced synthetic polymers for antibacterial orthopedic applications owing to their excellent biocompatibility and biodegradability. Zein is a biopolymer found in corn. The lacking mechanical stability of zein is overcome by incorporating bioceramics, e.g., mesoporous bioactive glass nanoparticles (MBGNs). In the present study, pure zein and zein/Zn–Mn MBGN composite coatings were deposited via electrophoretic deposition (EPD) on 316L stainless steel (SS). Zn and Mn were co-doped in MBGNs in order to make use of their antibacterial and osteogenic potential, respectively. A Taguchi design of experiment (DoE) study was established to evaluate the effect of various working parameters on the morphology of the coatings. It was observed that coatings deposited at 20 V for 5 min with 4 g/L concentration (conc.) of Zn–Mn MBGNs showed the highest deposition yield. Uniform coatings with highly dispersed MBGNs were obtained adopting these optimized parameters. Scanning electron microscopy (SEM), energy dispersive spectroscopy (EDS), X-ray diffraction (XRD), and Fourier transform infrared spectroscopy (FTIR) were employed to investigate the morphology and elemental composition of zein/Zn–Mn MBGN composite coatings. Surface properties, i.e., coating roughness and wettability analysis, concluded that composite coatings were appropriate for cell attachment and proliferation. For adhesion strength, various techniques, including a tape test, bend test, pencil hardness test, and tensile test, were performed. Wear and corrosion analysis highlighted the mechanical and chemical stability of the coatings. The colony forming unit (CFU) test showed that the zein/Zn–Mn MBGN composite coating was highly effective against *Staphylococcus aureus* (*S. aureus*) and *Escherichia coli* (*E. coli*) due to the presence of Zn. The formation of a hydroxyapatite (HA)-like structure upon immersion in the simulated body fluid (SBF) validated the in vitro bioactivity of the coating. Moreover, a WST-8 assay depicted that the MG-63 cells proliferate on the composite coating. It was concluded that the zein/Zn–Mn MBGN coating synthesized in this work can be used for bioactive and antibacterial orthopedic applications.

## 1. Introduction

The worldwide market of orthopedic implants is estimated to reach at USD 66,636 million by the year 2025 due to the increased rate of chronic diseases, trauma, and age factors [1].

Bio-implants based on metallic and ceramic composite materials provide certain properties, such as chemical stability and strength, along with good biocompatibility and bioactivity [2]. These features and increased demand encouraged the research in the orthopedic implants industry. Metallic implants exhibit relatively low corrosion resistance and uncontrolled leaching of toxic metal in a physiological environment. Released metal ions may cause tissue damage caused by carcinogens [3,4,5].

In this context, present studies are focused on inducing a positive response from body organs at the site of implantation [6,7,8]. The materials selected for substrates and coatings should be able to bond with soft and hard tissues and mimic the structure of natural bone. This requires the materials to be bioactive and form a hydroxyapatite (HA)-like structure on the surface of the implant [9,10]. The challenges in orthopedic implants are to overcome corrosion and toxic metal ion release, while improving bioactivity and resistance towards bacterial attack [11,12,13].

Zein is a natural biopolymer obtained from the endosperm of maize. It is a biocompatible and biodegradable material composed of prolamin [14]. In recent years, owing to its favorable properties, zein is being used as a coating material on implants, targeted drug delivery, and tissue engineering applications [15,16,17,18]. However, zein has a low mechanical strength and, for this reason, it is usually mixed with inorganic ceramic materials, such as silicate glasses, particularly, 45S5 bioactive glass (BG), hydroxyapatite (HA), and calcium phosphates [19,20,21]. These inorganic materials are bioactive; therefore, they promote bone formation. The brittle nature of ceramics is compensated by the ductility of zein, and the deficient mechanical stability of zein is overcome by the ceramics. Hence, the resulting composite performs better in both aspects, i.e., in biological as well as mechanical terms [20].

Moreover, it is important to consider the antibacterial effect because the formation of bacterial film over the surface of metallic implants is a major cause of infections [22,23]. To prevent this, the metallic ions (e.g., silver, copper, and zinc) are incorporated in biopolymer–ceramic composites. Furthermore, to enhance the osteogenic properties of implants, therapeutic ions, such as Mn and Sr, are added [24,25,26,27].

In general, polymeric materials for biomedical applications are heat sensitive and room temperature processing maintains their natural framework. Electrophoretic deposition (EPD) is a facile, cost-effective, and convenient deposition method operating at room temperature [28]. EPD has the advantage of producing uniform coatings over complex-shaped orthopedic implants of any size [29]. The suspension for EPD composed of biopolymers, ceramic particles, and metallic incorporations is prepared in the first step and deposited over the substrate of choice in the second step. The working parameters in the EPD process, such as voltage, time for deposition, concentration of the suspension, and temperature, can be easily altered according to the coating requirements [30]; however, it is a time-consuming and tedious job to perform the deposition by considering these factors one by one. The adaptation of the Taguchi design of experiment (DoE) approach minimizes the number of experiments. The data acquired from the experimental runs are analyzed through Minitab™ software [31]. It provides the optimum working conditions at which suitable coatings can be deposited [32,33,34].

Kaya et al. [35] deposited zein coatings on a stainless-steel (SS) substrate for the first time. Homogeneous coatings with a high deposition rate were obtained at various voltages and times, proving EPD a suitable method for the deposition of zein. In recent years, zein was deposited over Ti and Mg substrates other than SS for orthopedic applications [21,30]. However, SS still remains the first choice for short-term implantable devices due to the ease of availability and cost effectiveness. The focus of the ongoing research on SS implants is to enhance their biological properties while mitigating inherent flaws [36,37].

In the present work, a composite coating based on zein and mesoporous BG nanoparticles (MBGNs) is prepared for orthopedic applications. The co-deposition of zein and Zn–Mn MBGNs via EPD is investigated. The mesoporous structure of BG helps in augmented osteo-induction by providing an active surface area for osteoblast cells to attach and proliferate. Zn and Mn are added for antibacterial and osteogenic effects, respectively. The doping of metallic ions in MBGNs to produce a synergetic antimicrobial effect and osteogenesis is the most recent trend [38,39].

To the best of the author’s knowledge, the coatings comprising zein and Zn–Mn MBGNs have not been studied previously for orthopedic applications. Furthermore, the application of the Taguchi DoE approach to optimize zein/Zn–Mn MBGN coatings is being studied for the first time. Thus, the DoE approach will help in determining the significant factor (i.e., voltage, time, and conc.) affecting the quality of deposits, which in turn help us to understand the deposition kinetics of zein-based composite coatings.

Hence, this study contributes to the current literature available on antibacterial coatings with enhanced bioactivity and mechanical stability. The aim of this study is to study the suitability of mechanically stable zein/Zn–Mn MBGNs to inhibit biofilm growth and promote the bioactivity of the coatings.

## 2. Materials and Methods

### 2.1. Materials

Zein powder and acetic acid (99.8%) were purchased from Sigma-Aldrich (Taufkirchen, Germany) and VWR International (Shanghai, China), respectively. Zn–Mn-doped mesoporous bioactive glass nanoparticles (Zn–Mn-doped MBGNs) with nominal compositions of SiO_2_: 70 mol.%, CaO: 22 mol.%, Mn: 5 mol.%, and Zn: 3 mol.% were synthesized by the modified Stöber process [40]. The precursors, such as hexadecyltrimethylammonium bromide (CTAB)-98% (Sigma-Aldrich), zinc nitrate hexahydrate (Sigma-Aldrich), calcium nitrate (Avantor, Shanghai, China), manganese chloride (Uni-Chem, New Jersey, USA), tetraethyl orthosilicate (TEOS)-99% (Sigma-Aldrich), and ethyl acetate (99.5%, Merck, Darmstadt, Germany) were used to prepare MBGNs following the recipe obtained from our previous manuscript [41]. Briefly, a solution of distilled water (52 mL) and CTAB (1.12 g) was prepared by stirring (30 min) at room temperature. Subsequently, 16 mL of ethyl acetate and 52 mL of diluted ammonium hydroxide (32 vol.%) were added into the solution one by one. The pH was maintained at 9.6. Then, 12 mL of TEOS was added dropwise to the solution. Next, metallic precursors calcium nitrate and zinc nitrate were added, respectively. The solution was kept under continuous stirring the whole time. Then, the solution was left in a dry place for 4 h to allow the reaction between the reactants, and then centrifuged at 8000 rpm for 10 min to collect the particles. The drying was performed in an oven at 70 °C overnight. At the end, the calcination of dried particles was performed in a muffle furnace at 700 °C for 5 h.

The above synthesized Zn–Mn MBGNs were used in the zein suspension for composite coatings on AISI 316L stainless-steel (called 316L SS hereafter) substrates. Absolute ethanol (≥99.8%, VWR International) and distilled water (ELGADV 25 PURELAB option R7BP) were used to prepare the zein/Zn–Mn MBGNs suspension. Acetone (99.5%, Merck) was used to clean 316L SS before depositing the composite coatings.

### 2.2. Suspension Preparation

In order to deposit the composite coating, 316L SS having dimensions of 3 cm × 1.5 cm × 0.05 cm was used. To prepare the suspension for coating, 3 g of zein powder was added to the solution containing 3 mL of distilled water and 10 mL of acetic acid in a 50 mL beaker. Later on, 1, 2, 3, and 4 g/L Zn–Mn MBGNs were added to the zein mixture to prepare four different types of suspensions [19].

This mixture was magnetically stirred (SCI-LOGEX, MS-H280-Pro) for 30 min at room temperature to disperse the zein particles homogeneously. Then, 37 mL of absolute ethanol was added. The suspension was kept in the sonication bath for ~30 min in order to ensure the uniform dispersion of Zn–Mn MBGNs in the suspension, as reported in [42]. Finally, the suspension was magnetically stirred for 30 min. The pH of the suspension was measured by a Hanna HI98108 pH meter and maintained at ~3 by adding acetic acid drop-wise.

Before coating, 316L SS was ultrasonicated in 50 mL of mixture composed of 25 vol.% (12.5 mL) ethanol, 25 vol.% (12.5 mL) acetone, and 50 vol.% (25 mL) distilled water, and dried in air. An area of 2.25 cm^2^ was selected to deposit the coatings on 316L SS. Figure 1 shows a schematic representation of the preparation steps for the stable suspension of zein/Zn–Mn MBGNs and the subsequent deposition of composite coating on 316L SS.

Zein/Zn–Mn MBGNs suspensions were prepared with various concentrations of Zn–Mn MBGNs, as presented in Table 1.

The concentration of Zn–Mn MBGNs in the suspension was selected following the study conducted by Pawlik et al. [42]. It was shown that the higher the concentration of Zn–Mn MBGNs, the poorer the suspension stability. On the other hand, the <1 g/L concentration of Zn–Mn MBGNs led to poor bioactivity and mitigated antibacterial behavior. Thus, the concentration of 1–4 g/L of Zn–Mn MBGNs was selected in the current study.

### 2.3. Zeta Potential Measurement

The zeta potential of pure zein, Zn–Mn MBGNs, and zein/Zn–Mn MBGNs suspensions was measured in ethanol. The stability of the suspensions was evaluated using zetasizer (Malvern Instruments, Malvern, UK). Three measurements were obtained for each suspension and the average value is reported with a standard deviation.

### 2.4. Electrophoretic Deposition

In the present work, the stable suspension of zein/Zn–Mn MBGNs was prepared and then deposited on 316L SS (cathode) by EPD. The counter electrode (anode) of the same material and dimensions was used to complete the circuit. An inter-electrode separation during EPD was maintained at 10 mm. A constant current was applied by a DC power supply (eTM-605). Zein showed a positive zeta potential at pH~2.6 and moved towards the cathode upon the application of an electric field. It was hypothesized that Zn–Mn MBGNs were encapsulated by the positively charged zein molecules [43].

### 2.5. Taguchi Design of Experiment (DoE) Approach

In order to optimize the parameters and suspension composition for EPD, a Taguchi array of experiments was designed by using software (Minitab 17™, Beijing, China), which produced an array of 16 runs. Three control factors, applied voltage (A), deposition time (B), and conc. of Zn–Mn MBGNs in the zein suspension (C), were used to construct an orthogonal Taguchi DoE array. Four levels were assigned to each factor, as illustrated in Table 2.

The control factors were varied according to the selected levels and their effect on the deposition yield (mass of zein/Zn–Mn MBGN composite coating) was taken into consideration.

According to the DoE array, 16 experiments were performed, as shown in Table 3. Each experiment was repeated three times and the average value of deposition the yield, standard deviation, and signal-to-noise ratio (S/N) for deposition yield were calculated.

Equation (1) was used to calculate the deposition yield.
(1)$$\mathrm{Deposition}\;\mathrm{yield}=\frac{\Delta \;\mathrm{weight}}{A\;}\left(\frac{\mathrm{mg}}{{\mathrm{cm}}^{2}}\right)$$
where ∆ weight = weight after coating—weight before coating, while A = area of coating.

We aimed to obtain a higher deposition yield coating with a lower standard deviation value. The S/N ratio for the deposition yield was calculated by using Equation (2), considering that a higher deposition yield value results in the production of better coatings.
(2)$$\frac{S}{N}\;\mathrm{ratio}\;\mathrm{of}\;\mathrm{deposition}\;\mathrm{yield}=-10\;\log\left[\frac{1}{n}\left(\sum^{\text{​}}\frac{1}{y^{2}}\right)\right]$$
where *y* = deposition yield, *n* = no. of observations.

The ratio of *S*/*N* for the standard deviation was calculated using Equation (3) by considering that a lower standard deviation value results in better coatings.
(3)$$\frac{S}{N}\;\mathrm{ratio}\;\mathrm{of}\;\mathrm{deposition}\;\mathrm{yield}=-10\;\log\left[\frac{1}{n}\left(\sum^{\text{​}}y^{2}\right)\right]$$
where *y* = deposition yield, *n* = no. of observations.

A total of 48 experiments were performed. Each substrate before and after deposition was weighed by using a digital microbalance (Shimadzu-AUY220) (accurate up to 10 mg).

### 2.6. Characterization of Zein/Zn–Mn MBGN Composite Coating

#### 2.6.1. Morphological Analysis

The morphological analysis of the composite coating was conducted by using a high-resolution field emission scanning electron microscope (FE-SEM, MIRA, TESCAN). The thickness of the pure zein coating and zein/Zn–Mn MBGN composite coating was also examined. Cross-sections of the samples were prepared by grinding on sand paper (grit size 600) and then cleaned with a blower to remove any loosely bound particles on the edges. The qualitative compositional analysis of the composite coating was conducted by an energy-dispersive X-ray spectroscope (EDX) paired with SEM.

#### 2.6.2. Fourier Transformation Infrared (FTIR) Analysis

The composite coating was also assessed by the attenuated total reflection Fourier transformation infrared (ATR-FTIR) spectroscope (Thermofisher Nicolet Summit Pro) equipped with a Quest ATR unit (diamond crystal). The intensity spectrum was recorded in transmittance mode in the range of 4000–500 cm^−1^, with Happ-Genzel apodization, at 40 scans per spectrum and a resolution of 4 cm^−1^. The equipment was integrated with OMNIC paradigm software.

#### 2.6.3. Surface Roughness

The average and maximum surface roughness of the composite coating was evaluated by laser profilometer (UBM, ISC-2). A line length of 5–7 mm was drawn on the coated surface to measure the roughness of the surface of each sample (scanning velocity of 400 points per second). Bare 316L SS, zein-coated, and zein/Zn–Mn MBGN composite-coated samples were subjected to this test for comparison.

#### 2.6.4. Wettability Test

The contact angle (CA) was calculated by manually placing a 5 µL droplet of water on the surface of bare 316L SS and composite-coated sample with the help of a microliter pipette. The images of the droplet were obtained using a digital camera after 5 s of dropping. The contact angle between the surface and drop was measured by analyzing the images via ImageJ ™ software.

#### 2.6.5. Adhesion Tests

Tape test: a tape test was used to evaluate the qualitative adhesion strength of the composite coating deposited by the standard ASTM D3359-17, elaborated on in [44]. The results of the tape test were judged with an optical microscope (Novex, Arnhem, Holland). Small squares of 1 mm^2^ were marked on the coated surface by applying shear force to the coating. Strong adhesive tape was applied to the coating area and pressed. After 90 s, the tape was pulled off from the substrate. The adhesion strength of the composite coating was analyzed by the quantity of peeled-off coating from the substrate.

Tensile test: the tensile adhesion test was performed by the tensile pull-off method according to ASTM D4541-17 [45] and ISO 4624 [46]. The tensile pull-off test was performed by applying magic epoxy (made in Pakistan) between two bare surfaces of 316L SS (reference sample). Similarly, epoxy was applied between one coated and one non-coated surface. The dimensions of the samples (60 mm × 30 mm × 0.5 mm) were the same for both sets. After the epoxy dried, both sets of samples were subjected to tensile testing (HD-615A-S) by pulling the steel strips upon exerting a force perpendicular to the coated surface in an effort to remove the strips. The force at which the sample strips were detached from each other was recorded.

Pencil scratch test: a pencil scratch test according to the ASTM D3363-20 [47] was used for the evaluation of the adhesion strength of the composite coating. The coated samples were placed on a horizontal solid surface and tested by a set of pencils graded from hard (2H) to soft (8B). The sharpened stub of the pencil was placed at an angle of 45° against the coated surface. This test began with a pencil hardness of 2H and continued down the hardness range until the pencil could not scratch the coated surface anymore.

Bend test: Zein/Zn–Mn MBGN coatings were subjected to a bend test according to ASTM B571-97 [48]. This test determined the deformation ability of the composite coatings. The coated samples were bent at 180° with the help of tweezers to examine any kind of detachment, crack, or any distortion. The samples with the dimension of 60 mm × 30 mm × 0.5 mm were used for this purpose.

Wear test: a wear test was performed on pure zein and composite coatings using a Tribometer (MT/60/NI, Spain). The total sliding distance was kept at 50 m and a 1 N load was applied to the coatings with a 6 mm diameter stainless-steel ball indenter at 30 rpm. The wear test was performed under dry conditions at room temperature. All tests for adhesion were performed in triplicate and the average results are reported.

#### 2.6.6. Corrosion Behavior

The corrosion behavior of the coating and bare substrate was investigated using a potentiodynamic polarization scan. A Gamry instrument (reference 600) powered a three-electrode system with coated and uncoated samples as working electrodes, Ag/AgCl as a reference electrode, while graphite was used as a counter electrode. Initially, an open circuit potential (OCP) was determined for 30 min. All potentiodynamic polarization scans were recorded at 37 °C in simulated body fluid (SBF) electrolyte at a 2.5 mV/s scan rate in the potential range of ±500 mV. The corrosion potential (E_corr_) and corrosion current density (I_corr_) were measured directly by the extrapolation of the Tafel region. The tangents were drawn across the anodic and cathodic curves. At the intersection of the tangents, lines were drawn along the x and y axes to obtain the values of I_corr_ and E_corr_, respectively. The corrosion study was performed in triplicate for the coated and uncoated samples and the average results are reported here.

#### 2.6.7. Ion-Release Profile

Zn, Mn, Si, and Ca-ion release from synthesized MBGNs was determined by dispersing the MBGNs in SBF (75 mg/50 mL) for 21 days. The samples were placed in a shaking incubator at 37 °C to measure the release of ions under dynamic conditions using inductively coupled plasma-optical emission spectrometry (ICP-OES) (Thermo Scientific iCAP 6000). The aliquots were taken out at days 1, 3, 7, 14, and 21, and re-filled with fresh SBF.

#### 2.6.8. Antibacterial Analysis (Colony Forming Unit)

Zein/Zn–Mn MBGN-coated samples were subjected to an antibacterial test. The colony forming unit (CFU) method was used to check the bacterial inhibition potential of the coating. The coated sample (1 cm × 1 cm) was first incubated in the nutrient broth inoculated with *Staphylococcus aureus (S. aureus)* and *Escherichia Coli (E. coli*) separately at 37 °C in a shaking incubator. After 48 h, the supernatant (at 11,000 rpm for 10 min) was collected and further incubated with both bacteria at 37 °C for 24 h. A total of 50 µL aliquots of both solutions collected after serial dilutions were spread over agar plates and incubated in the same conditions. Finally, after 24 h, the number of colonies were counted. The control plates consisted of bacterial cultures without any addition of test samples.

#### 2.6.9. In Vitro Bioactivity

To investigate the in vitro bioactivity of coating in terms of hydroxyapatite (HA) formation, the SBF was prepared in the laboratory according to the method proposed by Kokubo et al. [49]. The coated samples with dimensions of 10 mm × 10 mm × 0.5 mm were immersed in 30 mL of SBF and then incubated at 37 °C for 1, 2, 3, 5, and 7 days. The pH was measured after 24 h for each sample and SBF was changed after every 2 days to maintain the ionic concentration of the test medium. At each time point, samples were removed from SBF, rinsed with de-ionized water, left to dry in air, and then stored in a desiccator. The weight of the samples was measured before and after treatment in SBF at each time point. The formation of HA was examined with SEM/EDX. For comparison, samples before immersion in SBF were also characterized.

#### 2.6.10. In Vitro Cellular Studies

The cytocompatibility of zein/Zn–Mn MBGN coatings was determined using a human osteosarcoma MG-63 cell line (Sigma Aldrich). 316 L SS and a tissue culture plate (TCP) were used as negative and positive controls, respectively. The direct assay method was conducted in which MG-63 cells were initially cultured in Dulbecco’s Modified Eagle’s Medium (DMEM) in 100 µg/mL of penicillin/streptomycin and 10 vol.% fetal bovine serum (FBS). Canted-neck cell culture flasks (Greiner Bio-One) were used to grow the cells. The culture was kept in a 5% CO_2_ incubator with a 95% humidity level at 37 °C for 48 h. Zein/Zn–Mn MBGN-coated samples were cut into 10 mm disks, placed in a 24-well plate, and sterilized under UV for 1 h. Later, 1 mL (10^5^ cells/mL) of cell culture was added to each well containing control and sample disks.

The well plate was then kept in a 5% CO_2_ incubator for 2 and 7 days at 37 °C. A WST-8 (water-soluble tetrazolium dye) kit was used to quantify cell viability. Furthermore, one-way ANOVA (significance level: *p* < 0.05, and *p* < 0.001) was adopted to statistically evaluate the data. The results were reported as the mean ± standard deviation (SD) of the experiment performed in triplicate. All the materials used for the cell-culture were purchased from Sigma-Aldrich, Taufkirchen, Germany. SEM was performed to analyze the attached cell morphology on control and sample disks. A total of 1 mL of 50% of glutaraldehyde solution (0.1%) was used to fix the cells on the surfaces.

## 3. Results and Discussion

### 3.1. Morphology of the Synthesized Zn–Mn MBGNs

The Zn–Mn MBGNs were synthesized via a modified Stöber process. The SEM of synthesized particles was performed to confirm the morphology and calculate the average particle size, as shown in Figure 2. The inset figure presents the calculated diameter of three different particles (Figure 2A). The average particle size of the as-synthesized Zn–Mn MBGNs was around 90–100 nm. The particles were round-shaped, porous, and well dispersed. The porous morphology of the particles was due to the Stöber process. In the physiological environment, circular particles have excellent flow properties. However, a minor morphological change in the as-synthesized particles does not affect the final properties of the coating [50,51]. The elemental composition of as-synthesized Zn–Mn MBGNs is shown in Figure 2B. The area analysis indicates the presence of Si, Ca, Zn, and Mn, thus confirming the doping of Zn and Mn in MBGNs.

### 3.2. Suspension Stability

The suspension stability was evaluated by measuring zeta potential of the suspensions. The zeta potential of the pure zein suspension was +27.6 ± 11.8 mV and Zn–Mn MBGNs had a value of 16.2 ± 3.8; whereas, for zein/Zn–Mn MBGNs suspension, it was +18.3 ± 6.7 mV. The zeta potential is lowered after adding Zn–Mn MBGNs to the zein suspension, which is due to the increased number of particles in the suspension [52]. The deposition was cathodic owing to the positive zeta potential of the zein molecules. The suspension containing zein and Zn–Mn MBGNs was stable enough to allow for the deposition for 5 min without particles settling down. The positive charge on the surface of Zn–Mn MBGNs and subsequent repulsive force (F_R_) among Zn–Mn MBGNs and cationic zein molecules contributed towards sufficient suspension stability.

Zeta potential is a function of pH and electrostatic forces of the interaction between suspended particles [52]. For the sufficient electrophoresis of particles, the zeta potential should exhibit a value of around +30 mV. Below this value, the particles will become unstable and agglomerate [53]. On the other hand, very high values of the zeta potential also result in mitigated electrophoretic mobility, the reason being that the electric field may not overcome the forces of repulsion between the particles rendering them unable to move in the direction of the applied field. The suspension stability is assured by the repulsive electrostatic forces between cationic zein molecules.

A zein molecule is chemically composed of amine (NH_2_) and carboxyl (COOH) groups [35]. It has both hydrophilic as well as hydrophobic domains. Due to the insolubility of zein in water, it is mostly referred to as hydrophobic [54]. Although zein is positively charged in the suspension, there remain some functional groups that are negatively charged, i.e., COO^−^. These charges may interact with oppositely charged (+ve) molecules and induce electrostatic forces of attraction (F_A_) between zein and Zn–Mn MBGNs, as illustrated in Figure 3. The F_A_ between zein and Zn–Mn MBGNs causes the polymeric chain to envelop the MBGNs. The zein-enveloped particles maintain repulsive forces among themselves in the suspension due to same cationic natures. Thus a state of stability is obtained in the suspension. Pishbin et al. [55] also proposed a co-deposition mechanism of chitosan and BG suspension by charge stabilization.

### 3.3. Deposition Mechanism of Zein/Zn–Mn Doped MBGNs

In an acidic environment, zein has a positive charge, due to which it moves towards a cathode. The pKa value (acidic dissociation constant) for zein is 18.6 (source: ChemAxon, Budapest, Hungary). The pKa value determines the solubility of the molecules in an acidic or basic medium. If the pH of the suspension is lower than its pKa value, then zein is soluble in solution and vice versa.

In zein, polar and non-polar functional groups are present. In the highly concentrated solutions of ethanol (>90%), the protonation of zein occurs, which leads to the deposition of zein molecules on the cathode. However, if the ethanol solution is less concentrated (<90%), then zein becomes negatively charged and migrates towards the anode. Hence, there is a possibility to manipulate the deposition of the zein molecules on either a anode or cathode [19,35,56]. As the ethanol used in this study is absolute (99%), a cathodic deposition occurs due to the protonation of zein molecules. Meyer et al. [57] suggested that the deposition of zein/BG composite particles on the cathode could be due to the bonding of the hydrophilic side-groups of zein with BG particles and their co-migration towards the cathode.

When the voltage is applied, a reduction reaction occurs near the cathode. Here, the local pH increases due to the production of hydrogen and hydroxyl ions. The amine group of zein is protonated in the aqueous medium (H_3_O^+^). The ammonium (NH_3_^+^) group of zein then migrates towards the negative electrode and reacts with hydroxyl ions at the cathode. As a result, the charge on the amine group of zein is lost and is deposited on the cathode along with Zn–Mn MBGNs. Sarkar et al. [58] also explained the deposition mechanism in consequence of charge neutralization.

### 3.4. EPD of Zein/Zn–Mn MBGNs (DoE Approach)

The best operating parameters for EPD and suspension concentration were obtained via the Taguchi DoE method. The method elucidates the maximum attainable deposition yield with a minimum standard deviation. For the coatings to be applied on orthopedic implants, it is important to maximize their bioactivity. The bioactive behavior of a coating material is the key parameter that determines the ability of an orthopedic implant to bond with soft and hard tissues [5]. Therefore, zein suspensions with high concentrations of Zn–Mn MBGNs will result in high deposition yields, thus increasing the bioactivity of the coating in a physiological environment. MBGNs form a hydroxyapatite (HA) layer over its surface upon contact with body fluid. The HA layer is bioactive and is known to induce osteoconductivity [59]. The co-doping of the Zn and Mn metallic ions produces a synergetic effect. Zn is proven to be an excellent antibacterial agent, thus it hinders the formation of biofilms on the implant [39,60]. Furthermore, Mn is an excellent osteogenic material widely used in bone tissue engineering [24,61].

Figure 4 shows the change in the deposition yield of zein/Zn–Mn MBGN coatings with respect to the levels of control factors. Figure 4A,B display the values of mean of means deposition yield and mean S/N deposition yield. The maximum values are obtained at A4 (20 V), B4 (7 min), and C4 (4 g/L of Zn–Mn MBGNs) in both cases. This confirms that there is no notable difference in the response of mean deposition yield and mean S/N deposition yield values. The deposition rate increases with the increase in applied voltage, time, and conc. of Zn–Mn MBGNs in the zein suspension. This is in complete agreement with Hamaker’s law [52].

In terms of the standard deviation, the minimum values are obtained at A4 (20 V), B3 (5 min), and C4 (4 g/L of Zn–Mn MBGNs), as shown in Figure 4C. Figure 4D shows that the maximum values of the mean S/N ratio for the standard deviation are A1 (5 V), B1 (1 min), C1 (1 g/L of Zn–Mn MBGNs). Thus, the best parameters in terms of standard deviation are A4, B3, and C4.

The parameters selected from the standard deviation values are comparable with the mean deposition yield values. Therefore, there is a need to compromise on certain parameters. Since these coatings require high deposition yields, preference is given to the coatings that are deposited with a high conc. of Zn–Mn MBGNs. According to the maximum–minimum values presented in Table 4 for the S/N response for deposition yield, voltage is the most effective factor to determine the deposition yield. In case of an S/N response to standard deviation, the maximum–minimum values shown in Table 5 infer that the conc. of Zn–Mn MBGNs is most effective parameter. For the parameter B4 (7 min), the standard deviation and S/N response to the standard deviation for the deposition yield are very high. Therefore, it is concluded that the best parameters are A4 (20 V), B3 (5 min), and C4 (conc. of Zn–Mn-doped MBGNs: 4 g/L of Zn–Mn MBGNs).

### 3.5. Morphological Analysis of Zein/Zn–Mn MBGN Coatings

The coatings were prepared using optimized EPD parameters (20 V, 5 min and 4 g/L Zn–Mn MBGNs) obtained by the Taguchi DoE approach. At first, the coatings were analyzed using an optical microscope. The optical microscope showed a uniform covering of 316L SS substrate by the zein/Zn–Mn MBGN coating. For an in-depth understanding of the surface morphology, SEM analysis was performed. Figure 5A,B show the SEM micrographs at low and high magnifications, respectively. It can clearly be observed that a homogeneous coating of zein/Zn–Mn MBGNs was produced at 20 V, 5 min, and 4 g/L conc. of Zn–Mn MBGNs. Zn–Mn MBGNs were uniformly dispersed throughout the zein matrix, as shown in Figure 5A. However, at some points, few Zn–Mn MBGNs were found in the form of agglomerates. A cross-sectional SEM image (Figure 5C) of the coating shows that the deposited film had a uniform thickness of ≈18 µm. Figure 5D shows the elemental mapping of the coating from the cross-section. The red-and-green-colored area is attributed to Fe and Cr representing 316L SS; whereas purple, orange, and dark green represent the constituents of the composite coating, i.e., Si, Ca, and C, respectively. EDS analysis was also performed to qualitatively determine the composition of the composite coating. The presence of Si, Ca, Zn, and Mn was evident from the EDS spectrum, as shown in Figure 5E. The elemental mapping validated the EDS spectrum. For further affirmation of the presence of zein/Zn–Mn MBGNs in the composite coating, FTIR was performed.

### 3.6. FTIR Analysis of Zein/Zn–Mn MBGN Coating

FTIR analysis monitors the interaction of molecules during EPD. It also evaluates the conformational variations occurring during the deposition process [33]. Amide groups (I, II, and III) of amino acids can be easily detected by FTIR.

Figure 6 shows the FTIR spectrum of zein/Zn–Mn MBGN composite coatings in the range of 2000–500 cm^–1^. The characteristic peaks of zein and Zn–Mn MBGNs were determined, as shown in Table 6.

Pure zein represented characteristics peaks of amide I (C=O), II (N-H), and III (C-N) at 1650 cm^–1^, 1510 cm^–1^, and 1230 cm^–1^, respectively. After incorporating Zn–Mn MBGNs in zein, a similar pattern was observed in the FTIR spectrum. The peaks attributed to the MBGNs were prominent at 800 cm^–1^ (Si-O-Si bending vibration), 1040 cm^–1^ (SiO_2_ stretching vibration), and 1067 cm^–1^ (Si-O-Si bending vibration). The aqueous medium of suspension causes Zn–Mn MBGNs to form a free hydroxyl group on its surface, which creates the possibility of H-bonding between the two materials [33,64]. The decreased intensity of C=O vibrations after adding Zn–Mn MBGNs to zein indicates H-bonding, due to which the co-deposition of zein and Zn–Mn MBGNs was possible to attain. The strong H-bonding of the composite coating also resulted in excellent adhesion strength (discussed in the subsequent sections).

### 3.7. Surface Roughness Test

Surface roughness is an important factor to consider for microbial and cell attachment. It greatly affects the performance of an implant material as the surface is the first thing that comes into contact with body fluids. According to the literature, osteoblasts attach to a surface having a roughness range of 1–2 µm [65]. In the present study, R_a_ for the composite coating was higher (1.8 µm) than the pure zein (1.2 µm) coating. This could be due to the incorporation of Zn–Mn MBGNs in the zein suspension. Our previous studies also confirm the increased R_a_ value after adding MBGNs to polymer coatings [24,66].

### 3.8. Wettability Test for Zein/Zn–Mn MBGN Coating

The wetting property of a coating determines its ability to facilitate protein adsorption on the surface. In an ideal situation, the surface should not be super hydrophilic or super hydrophobic. A CA range of 35–80° is considered suitable for excellent cell adhesion and proliferation [30]. Water CA was measured for 316L SS, pure zein coating, and zein/Zn–Mn MBGN composite coating. The 316L SS presented a CA value of 58 ± 6°. Pure zein showed an average CA value of 63 ± 4°, and for the composite coating it was measured as 80 ± 9°. The CA for pure zein was slightly higher than that of 316L SS. As the chemical structure of zein (shown in Figure 3) depicts that it has both hydrophilic and hydrophobic side-groups, it is quite possible that hydrophilic groups of zein were oriented towards the top of the surface during deposition, offering more attraction to the water drop. The CA of the composite coating was even higher than 316L SS and the pure zein coating. Although the addition of MBGNs alters the wettability of the surface, this value is still in the suitable range for protein adsorption and spreading. Similar results were reported by Mariotti et al. [67]. An increment in the CA value was observed after incorporating Cu-doped BGs in the zein fiber mat.

### 3.9. Adhesion of Zein/Zn–Mn MBGN Coating

A tape adhesion test was performed on the composite coating to assess the response of coating towards mechanical stresses. The composite coating was inspected under an optical microscope. Figure 7 shows the optical images of the composite coating before (Figure 7A) and after (Figure 7B) removing the tape. The composite coating clearly showed very little (<5%) delamination after removing the tape. Considering these results, the adhesion strength was graded as ‘4B’ according to the ASTM standard.

A tensile test for composite coatings was performed according to ASTM D3039 [68]. 316L SS rectangular plates joined with epoxy were detached at 55 MPa (Figure 7C), whereas the set of uncoated and coated plates showed detachment at 46 MPa (Figure 7D). It was observed that 316L SS had good adhesion with epoxy, thus the epoxy film was split on both sheets. However, the epoxy was detached from the composite coating side for the other set. A small amount of epoxy was still adhered to coated sheet 2, whereas no sign of detached composite coating was visible on epoxy. This test concluded that the composite coating was well adhered with the substrate.

To further analyze the scratch resistance of coatings, a pencil hardness test was performed. The scratching began with a hard-grade (2H) pencil and was continued down towards softer grades on the hardness scale (2H–8B). The samples were subjected to the test until the pencil could no longer scratch the coating. For each type of coating, the test was performed on three samples and the results were compared with the ASTM standard. It was revealed that the pure zein coating had a hardness grade of ‘7B’ (soft), whereas the composite coating showed a hardness of ‘4B’ (hard). The higher hardness value of the composite coating was attributed to the strong adhesion of the coating with the substrate.

The coatings were also subjected to a 180° bend, manually, as shown in Figure 7E,F. It was observed that both coatings were strongly adhered to the substrate and no sign of delamination was observed. The coatings were rated ‘4B’ according to the ASTM standard.

To realize a successful implantation in the physical environment, it is necessary to examine the wear behavior of coatings. Wear resistance also infers how an implant material reacts towards a somewhat vigorous handling of the implant during surgical procedures. To confirm the suitability of the composite coating, a wear study was performed. Figure 8 shows the partial wear tracks of pure zein (Figure 8a) and the composite coating (Figure 8b) observed under an optical microscope. The pure zein coating was removed almost completely after the test, whereas the composite coating was still intact at some points. The coefficient of friction (CoF) throughout the sliding distance was the same for both coatings (Figure 8A,B). Figure 8C shows the wear rate of both types of coatings. It is evident that the pure zein coating has a higher wear rate compared to the composite coating. The better wear resistance of the composite coating could be due to the presence of MBGNs, owing to the fact that BG belongs to the class of rigid bioceramics. This affirms their suitability for implant coatings.

### 3.10. Corrosion Behavior of Zein/Zn–Mn MBGN Coating

The potentiodynamic polarization curves of bare 316L SS and zein/Zn–Mn MBGN-coated samples are presented in Figure 9. It is a well-known fact that the corrosion test results are difficult to reproduce due to variations in the testing environment. Even though the test is standardized, the testing arrangement (apparatus) and manual handling contribute to the collected data [69]. Hence, the results reported here are an average of the experiments performed thrice.

The OCP for bare SS varied rapidly from −0.22 V to −0.19 V and then remained almost constant at −0.13 V up to 30 min; for the composite coating, OCP varied from −0.25 V to −0.21 V before becoming stable at the latter value. After OCP was stabilized, the polarization curves were obtained. The anodic polarization curve depicts the dissolution of iron and other alloying elements from 316L SS into respective ions. The cathodic branch of the polarization curve shows the hydrogen evolution. The junction of anodic and cathodic branches shifted towards the left in the case of the coated sample, which indicates a decrease in the I_corr_ value. The bare sample shows a more abrupt, significant increase in corrosion potential in the passive region compared to the coated sample, which is attributed to the rapid dissolution of the substrate. The measured E_corr_ and I_corr_ values are presented in the inset of Figure 9. It shows that the I_corr_ value for cthe omposite coating decreased by ~100 turns, which is a good exhibition of enhanced corrosion resistance. Therefore, the composite coating can improve the corrosion resistance of 316L SS.

In the graph, it can be observed that the composite coating has a lower pitting potential compared to the bare sample. This trend is common for polymeric coatings because of porosity, which is desirable in biological coatings. The presence of porosity allows the penetration of SBF towards the metal surface causing localized corrosion at the exposed surface. However, a detailed corrosion analysis with a long immersion time is required to completely understand the behavior of coated samples. In the future, electrochemical impedance spectroscopy (EIS) could be performed to evaluate the effect of corrosion caused by pitting.

### 3.11. Ion Release from Zn–Mn MBGNs

The ion-release profile of Zn–Mn MBGNs influences the antibacterial behavior, bioactivity, and cell biology directly. Hence, the Zn–Mn MBGNs synthesized in this study were analyzed in terms of ion release up to 21 days in SBF. Figure 10 shows the release of doped metallic ions: Zn and Mn, as well as Si and Ca ions (major constituents of bioglass). It is evident that Si and Ca initially release rapidly and then follow a relatively slow release rate. The Ca ion enhances the osteoconductive behavior of the MBGNs [41].

Furthermore, it was observed that Zn and Mn ions followed a linear release pattern in the specified time period, hence verifying a sustained release. However, the Mn ion had a high release rate from MBGNs compared to the Zn ion.

Zn and Mn ions play a major role in the physiological environment. Zn is a known antibacterial agent and may prevent the formation of biofilm on the coated surface. Mn augments the ostegenic potential of MBGNs; hence, the sustained release of Mn is helpful in the in vitro bioactivity of the coating.

### 3.12. Antibacterial Analysis (CFU)

A zein/Zn–Mn MBGN-coated sample was tested against *S. aureus* and *E. coli*. The bacterial growth on control plates was the maximum, as shown in Figure 11A,B. The coating effectively inhibited the growth of bacteria. It was observed that the coating fully inhibited the growth of *S. aureus* and zero CFU was attained (Figure 11C), whereas a partial inhibition was observed in the case of *E. coli* (Figure 11D). However, the colonies were in a countable range. Bacterial inhibition is attributed to the release of Zn from MBGNs, which is a strong antibacterial agent. The antibacterial effect also affirms the release of the Zn ion in the solution, as indicated in the Zn-ion-release profile (Figure 10). The dissolution products of MBGNs can also cause the death of bacteria via cell damage due to the increased pH [70].

### 3.13. *In Vitro* Bioactivity Test for Zein/Zn–Mn MBGN Coating

Bioactivity is an important aspect to consider while designing coatings for bone tissue engineering (BTE). Calcium phosphate (CaP) is a major constituent of human bone, and the materials able to produce a CaP-based interface (HA layer) between the implant and surrounding tissues are considered biologically active. An in vitro bioactivity test was performed for the zein/Zn–Mn MBGN composite coating for 1, 2, 3, 5, and 7 days of incubation. The SEM/EDX analysis performed on day 3 confirmed the formation of a cauliflower-like HA structure on the surface of the coating, as shown in Figure 12. The formation of a HA layer is an indicator of the bioactivity of the composite coating facilitating the osteo-induction as well as osteoconduction of coated implants. It was observed that HA started to form on day 3 (Figure 12A) as the composition of BG used in this study was composed of SiO_2_ and CaO in the ratio of 70:30 without any addition of phosphorous (P). The P is usually present in the form of amorphous calcium orthophosphate. It regulates the formation and proliferation of osteoblasts while hindering osteoclast differentiation. Hence, P may be considered an accelerator in bioactivity, and the delayed formation of HA could be due to absence of P in the BG [71]. Another possible reason for the delayed HA formation could be the release of Zn ions from the coating. Zn may become attached to the active growth sites of HA and retards the nucleation of HA in SBF [51]. However, HA was formed, which proved the bioactivity of the composite coating. This effect may be attributed to Ca-ion release from the MBGNs present in the coating, as indicated in Figure 10. The Ca ion is said to improve the bioactivity of the surfaces [72]. The EDX spectra (Figure 12B) also suggested the formation of a HA layer consisting of calcium phosphate on the surface of the coating.

### 3.14. *In Vitro* Cytocompatibility Test for Zein/Zn–Mn MBGN Coating

The cytocompatibility of zein/Zn–Mn MBGN composite coatings was investigated in MG-63 cells by WST-8 assay. Figure 13A shows that the zein/Zn–Mn MBGN coating supported the proliferation of MG-63 cells over the duration of 2 and 7 days compared to 316L SS. The difference in % cell viability for 316L SS, TCP, and the composite coating was significant at *p* < 0.05. After 7 days, there was a notable increase in the cell viability (~80%) of the coated sample. These results depict that the composite coating was cytocompatible with MG-63 cells, even after an extended period of contact. On the other hand, 316L SS is likely to release toxic ions, i.e., Ni and Cr, due to which cell death may occur. However, the composite coating inhibited the release of metallic ions, even after 7 days of incubation.

The results of cytocompatibility can be related to the surface roughness and wettability of the composite coating. The R_a_ of the composite coating (1.8 µm) was in the range of the reported value (1–2 µm) suitable for osteoblast attachment. The hydrophilic behavior of the coating also facilitated cell adhesion and proliferation [73,74]. Figure 13B,C show the spreading of the cell on the surface of the composite coating after 2 and 7 days of incubation, respectively. The morphology depicts that the cells grow with the increase in the incubation time. The MG 63 cells became attached to the surface and elongated with guided morphology. It can be concluded that the release of the Mn ion from the MBGNs upregulates the osteogenic marker genes, and extracellular matrix formation is enhanced in the presence of Mn [61]. Therefore, the loading of Mn in MBGNs is an effective technique to improve bone tissue engineering.

## 4. Conclusions

The following conclusions were drawn from the present study.

Zein/Zn–Mn MBGN composite coatings were deposited via EPD on 316L SS.The optimized parameters were obtained from the Taguchi DoE approach by running an array of experiments. The experiments deduced that coatings deposited at 20 V for 5 min with 4 g/L conc. of Zn–Mn MBGNs in zein suspension resulted in a high deposition yield.SEM/EDX analysis confirmed a uniform composite coating of thickness ≈18 µm.FTIR verified the hydrogen bonding of the zein polymer with Zn–Mn MBGNs, which resulted in the augmented adhesion strength of the composite coating with substrate.The surface roughness and wettability analysis affirmed the possibility of cell attachment and growth on composite coatings.For adhesion, the tape test, bend test, pencil hardness test, tensile test, and wear test were performed. The tests displayed favorable results according to ASTM standards.Corrosion behavior analysis of composite coatings showed that appreciable corrosion resistance was achieved.Ion-release profile confirmed the release of Si, Ca, Zn, and Mn ions from the MBGNs, which impart the required biocompatible, bioactive, antibacterial, and osteogenic properties to the composite coating.CFU test revealed the efficacy of Zn–Mn MBGNs in the zein coating against *S. aureus* and *E. coli*.Zein/Zn–Mn MBGN composite coatings formed a hydroxyapatite structure in SBF, proving the osteogenic potential of the composite coatings in bone tissue engineering.

The composite coating showed no cytotoxicity against MG-63 cells and the presence of Mn enhanced cellular attachment.

## Figures and Tables

**Figure 1 jfb-13-00097-f001:**
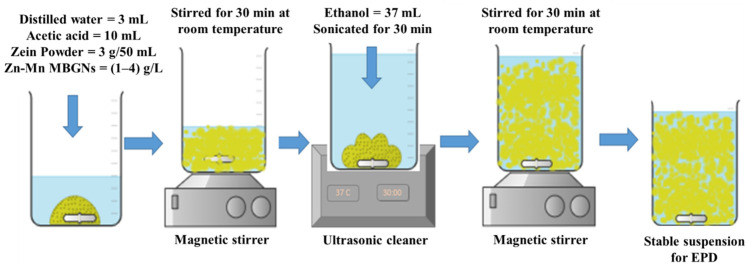
Schematic representation for the preparation steps of zein/Zn–Mn MBGNs stable suspension to deposit the composite coating on 316L SS.

**Figure 2 jfb-13-00097-f002:**
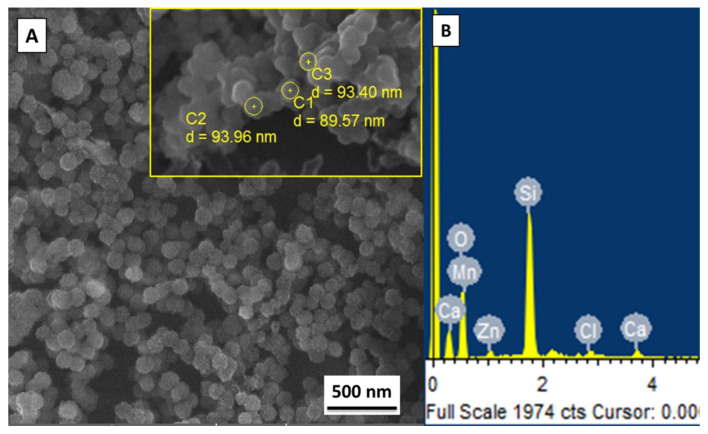
(**A**) SEM image showing the morphology and size of as-synthesized Zn–Mn MBGNs; (**B**) EDX spectrum confirms the presence of Zn, Mn, Ca, and Si.

**Figure 3 jfb-13-00097-f003:**
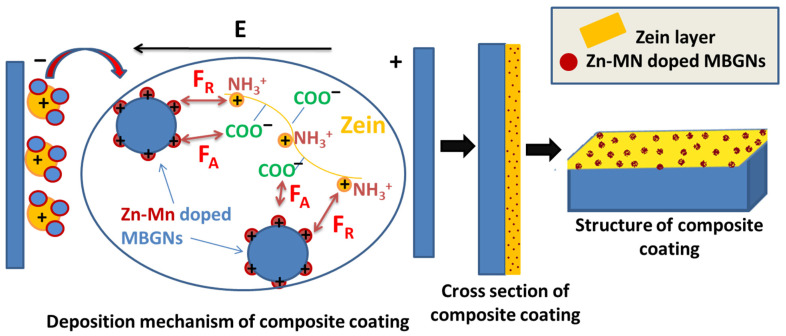
Deposition and stabilization mechanism of zein/Zn–Mn MBGNs suspension.

**Figure 4 jfb-13-00097-f004:**
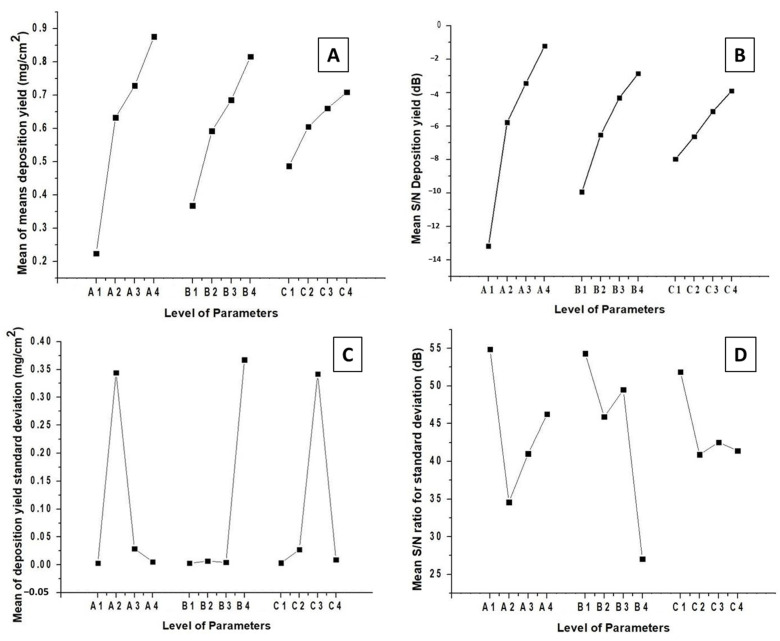
Effect of control parameters on the deposition yield of zein/Zn–Mn MBGN coating produced by EPD: (**A**) mean of means deposition yield, (**B**) mean S/N deposition yield, (**C**) mean of deposition yield standard deviation, and (**D**) mean S/N ratio for standard deviation.

**Figure 5 jfb-13-00097-f005:**
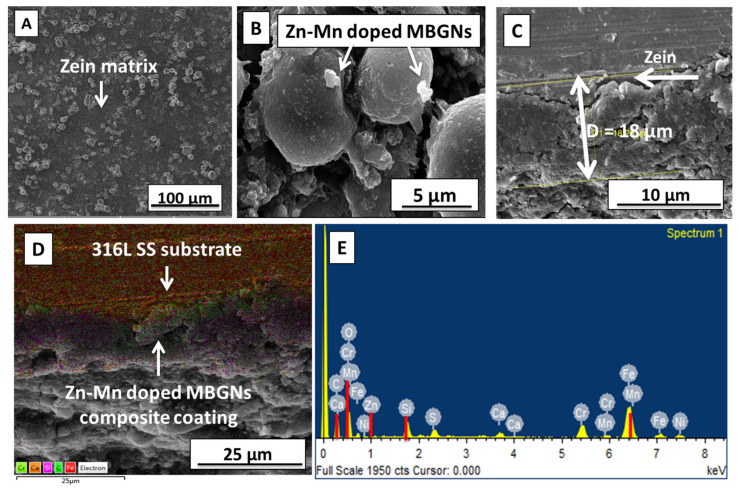
SEM images of zein/Zn–Mn MBGN coatings produced at 20 V, 5 min; (**A**) low magnification showing uniform deposition of the coating; (**B**) high magnification showing distribution of Zn–Mn MBGNs on zein particles; (**C**) cross-section; (**D**) elemental mapping; and (**E**) EDS spectrum of composite coating.

**Figure 6 jfb-13-00097-f006:**
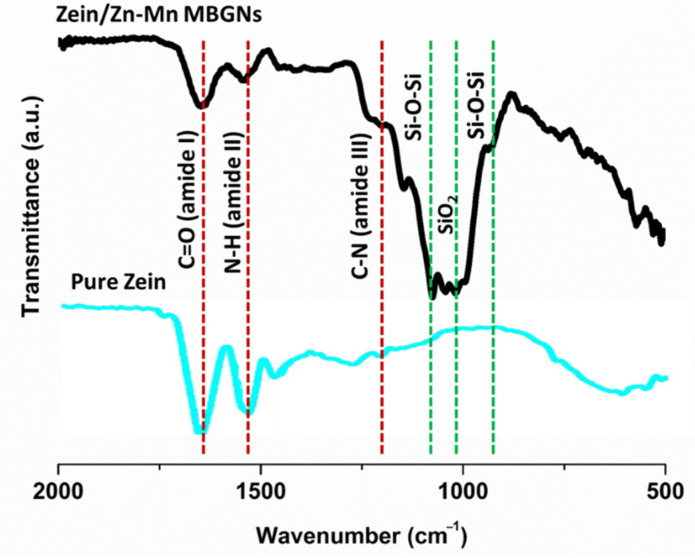
FTIR spectrum of pure zein and composite coating.

**Figure 7 jfb-13-00097-f007:**
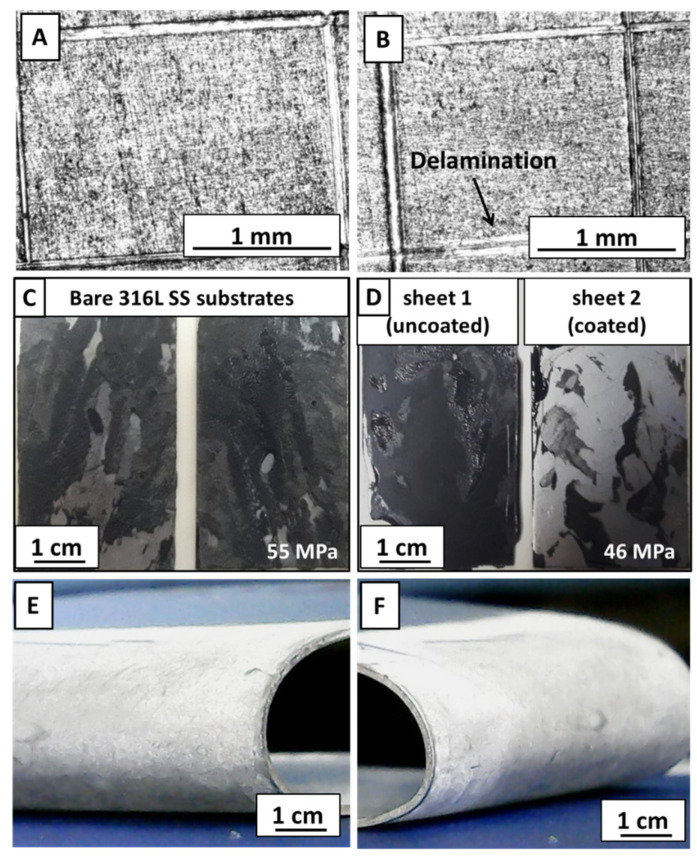
Tape test on composite coating: (**A**) before and (**B**) after removing tape; tensile test: (**C**) detached, bare 316L SS sheets and (**D**) bare 316L SS sheet 1 detached from coated sheet 2; bend test: (**E**) pure zein coating; and (**F**) composite coating without any cracks.

**Figure 8 jfb-13-00097-f008:**
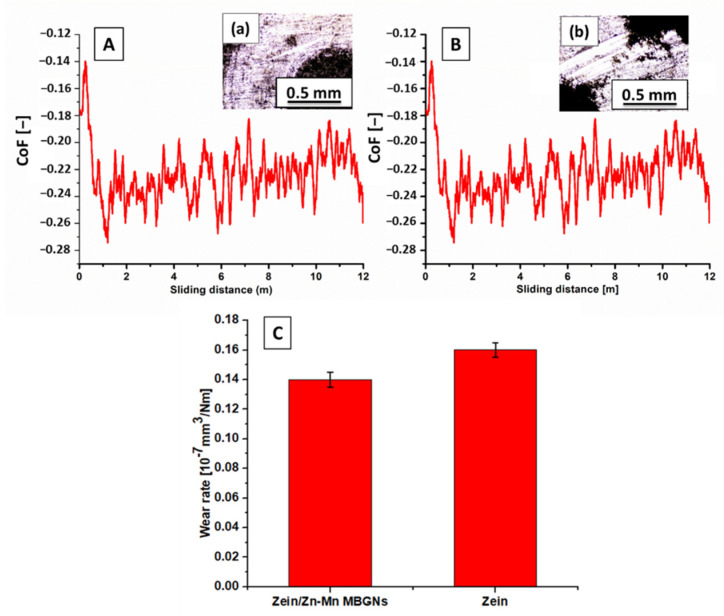
Graphs showing the CoF vs. sliding distance curves for (**A**) pure zein coating, (**B**) composite coating, and (**C**) wear rate of pure zein and composite coatings. Wear track images obtained with an optical microscope: (a) pure zein and (b) composite coating.

**Figure 9 jfb-13-00097-f009:**
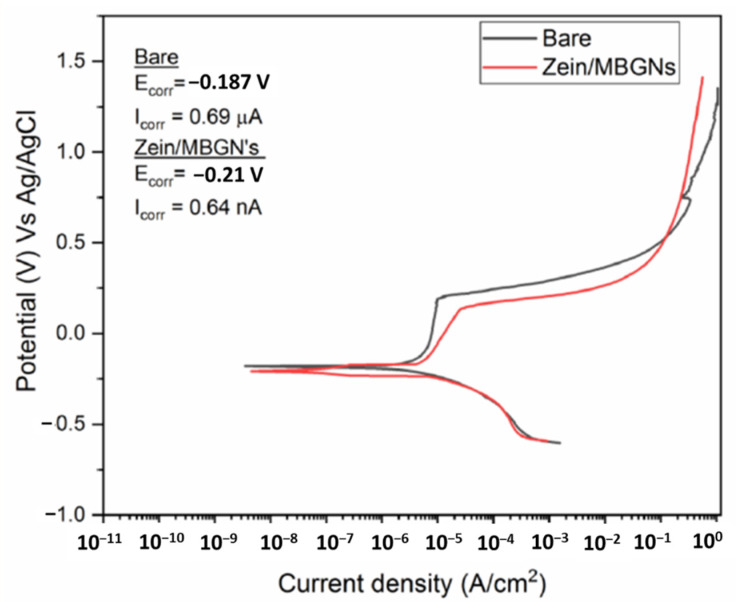
Potentiodynamic curve for base 316L SS and zein/Zn–Mn MBGN coatings.

**Figure 10 jfb-13-00097-f010:**
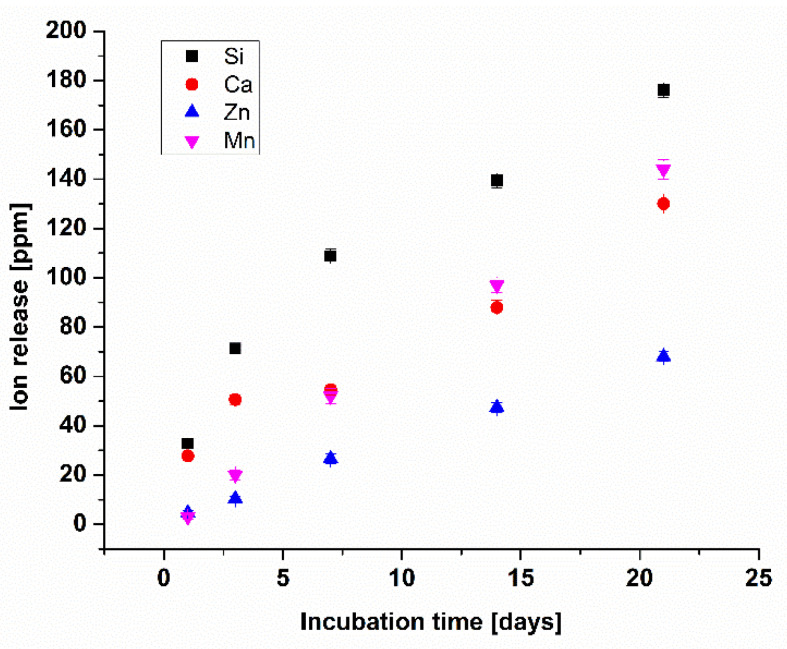
Ion-release profile of Zn–Mn MBGNs in SBF up to 21 days using ICP-OES. (Experiment was performed in triplicate and the mean values are reported with the standard deviation represented by the error bars in the figure).

**Figure 11 jfb-13-00097-f011:**
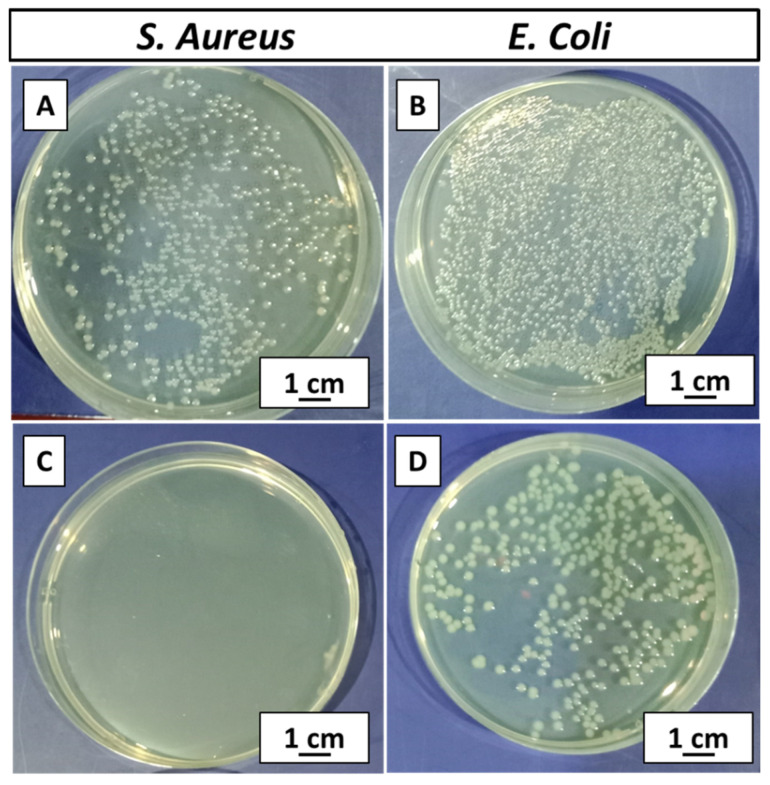
CFU results for zein/Zn–Mn MBGN composite coating, (**A**) control (*S. aureus*), (**B**) control (*E. coli*), (**C**) complete inhibition of *S. aureus*, and (**D**) partial inhibition of *E. coli*.

**Figure 12 jfb-13-00097-f012:**
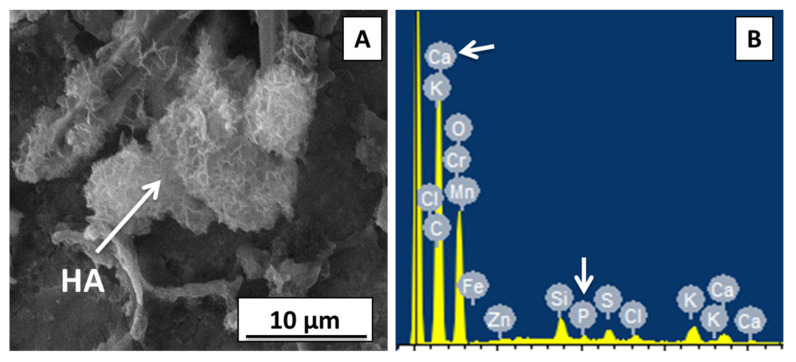
SEM/EDX analysis of composite coating after treating in SBF for 3 days: (**A**) formation of cauliflower-like HA structure, (**B**) detection of Ca and P confirms the presence of HA.

**Figure 13 jfb-13-00097-f013:**
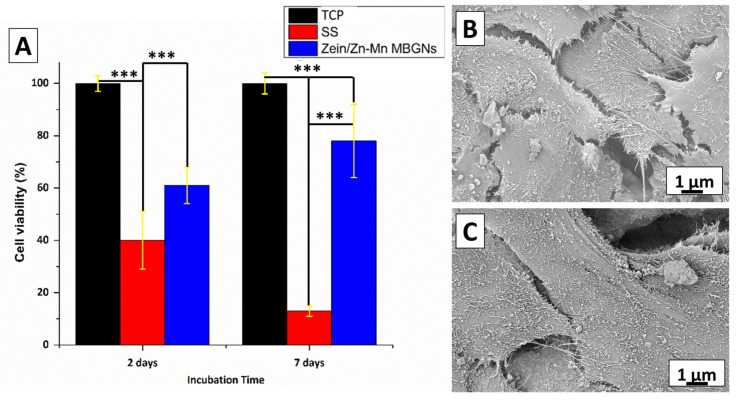
(**A**) Response of MG-63 cells towards zein/Zn–Mn MBGN coating up to 7 days of culture, *** symbolizes the significant difference between the two systems at *p* < 0.05, (**B**,**C**) SEM analysis of MG 63 cell morphology spreading on the surface of zein/Zn–Mn MBGN coatings incubated for 2 and 7 days in cultivation medium, respectively.

**Table 1 jfb-13-00097-t001:** Composition of zein/Zn–Mn MBGNs suspensions used to deposit composite coating.

Types of Suspensions	Composition				
	Zn-Mn MBGNs (g/L)	Zein (g/50 mL)	Acetic Acid (mL/50 mL)	Ethanol (mL/50 mL)	Distilled Water (mL/50 mL)
1	1	3	10	37	3
2	2	3	10	37	3
3	3	3	10	37	3
4	4	3	10	37	3

**Table 2 jfb-13-00097-t002:** Control factors along with four levels of deposition parameters.

Symbol	Control Factors	Levels			
		1	2	3	4
A	Voltage (V)	5	10	15	20
B	Time (min)	1	3	5	7
C	Conc. of Zn–Mn MBGNs (g/L)	1	2	3	4

**Table 3 jfb-13-00097-t003:** Experimentally measured values of deposition yield and the corresponding standard deviation, S/N ratio for the deposition yield, and S/N ratio for standard deviation for EPD of zein/Zn–Mn MBGN composite coating.

Run	Control Factors			Deposition Yield in mg/cm^2^	S/N Ratio of Deposition Yield (dB)	Standard Deviation	S/N Ratio of Standard Deviation (dB)
	Voltage (V)	Time (min)	MBGNs Conc. (g/L)				
1	5	1	1	0.17	−15.56	0.001	65.1485
2	5	3	2	0.20	−14.11	0.003	50.6404
3	5	5	3	0.26	−11.77	0.001	61.8456
4	5	7	4	0.27	−11.28	0.008	41.9239
5	10	1	2	0.25	−12.15	0.003	50.8454
6	10	3	1	0.28	−10.97	0.006	44.7428
7	10	5	4	0.96	−0.36	0.005	45.3928
8	10	7	3	1.04	0.34	1.363	−2.6900
9	15	1	3	0.36	−8.94	0.001	58.7529
10	15	3	4	0.90	−0.90	0.016	35.8934
11	15	5	1	0.61	−4.41	0.004	49.0038
12	15	7	2	1.05	0.46	0.095	20.4046
13	20	1	4	0.70	−3.08	0.008	42.4813
14	20	3	3	0.99	−0.12	0.002	52.2514
15	20	5	2	0.92	−0.73	0.008	41.6981
16	20	7	1	0.90	−0.96	0.004	48.5342

**Table 4 jfb-13-00097-t004:** S/N (dB) response for deposition yield of zein/Zn–Mn MBGN coating.

Factors	Level 1	Level 2	Level 3	Level 4	Δ = Max.–Min.
Voltage (V)	−13.181	−5.785	−3.450	−1.223	11.958
Time (min)	−9.934	−6.527	−4.316	−2.862	7.072
Conc. of Zn–Mn MBGNs (g/L)	−7.977	−6.632	−5.124	−3.906	4.071

**Table 5 jfb-13-00097-t005:** S/N (dB) response to standard deviation for deposition yield of zein/Zn–Mn MBGN coating.

Factors	Level 1	Level 2	Level 3	Level 4	Δ = Max.–Min.
Voltage (V)	54.89	34.57	41.01	46.24	20.32
Time (min)	54.31	45.88	49.49	27.04	27.26
Conc. of Zn–Mn MBGNs (g/L)	51.86	40.90	42.54	41.42	10.96

**Table 6 jfb-13-00097-t006:** Characteristic FTIR peaks of pure zein and composite coating.

Wave Numbers (cm^–1^)	Associated Bonds	Material	References
800	Si-O-Si	Zn-Mn MBGNs	[57,62]
1040	SiO_2_	Zn-Mn MBGNs	[57,62]
1067	Si-O-Si	Zn-Mn MBGNs	[62]
1230	C-N	Zein	[57,62]
1510	N-H	Zein	[57,62,63]
1650	C=O	Zein	[57,62,63]
2919	C-H	Zein	[57,62,63]
3292	O-H	Zein	[57,62,63]

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
