# Peer review of "Zn–Mn-Doped Mesoporous Bioactive Glass Nanoparticle-Loaded Zein Coatings for Bioactive and Antibacterial Orthopedic Implants"

_jfb, 2022, doi:10.3390/jfb13030097_

Round 1
Reviewer 1 Report
Dear Authors, here are my comments for this paper:
1- please add details about the preparation of Zn-Mn MBGNs (lines 110-115)
2- the description of volumes in line 124 and figure 1 is not in agreement with data in table 1. Furthermore, I suggest to indicate always the mL used and not vol% (scale-up results are unknown). Similarly, please write 6 g in line 123 (wt% as no meaning), and 37 mL in line 128.
3- please revise formula (1), line 181, A is in cm2 (deposition yield is in mg/cm2)
4- in line 294, what do Authors mean with "flow properties"?
5- line 304 and following- (a) the positive charge value on the surface of ZnMn MBGNs should be reported; (b) The zeta-potential of the suspensions reported indicate that they have low colloid stability. Obviously, the non-stable particles will sediment rapidly. Please comment in the text how you are able to obtain suspensions; (c) why the zeta potential gets lower by adding positive Zn-Mn MBGNs?
6- line 405- thickness of 18 microns sounds very high for achieving good mechanical properties.
7- FTIR analysis is not convincing: (a) the bands at 800, 1040 and 1067 cm-1 attributed to Zn-MBGNs are present also in bare zein, figure 6; (b) lines 431-432 are not supported by data: please discuss why hydrogen bonding
8- lines 445-446 region 35-80 degrees is fully hydrophilic, please change the sentence
9- line 455, please change "augments the hydrophobic character of composite coating" into "alters the wettability of the surface"
10- paragraph 3.11 is poor: ionic release fo Zn could support the discussion in line 525-526. Furthermore the release of Mn and of the metals from the metallic support should be measured for better understanding biocompatibility of these materials.
11- lines 534-536- the discussion about the presence of P should be supported by a suitable reference
12- in conclusions, point 4 is not supported by results.
Author Response
Reviewer 1
1- Please add details about the preparation of Zn-Mn MBGNs (lines 110-115)
Ans: The detail of preparation of Zn-Mn MBGNs (lines 117-126) is added as following.
“Briefly, a solution of distilled water (52 mL) and CTAB (1.12 g) was prepared by stirring (30 min) on room temperature. Afterwards, 16 mL ethyl acetate and 52 mL diluted ammonium hydroxide (32 vol.%) were added into the solution one by one. The pH was maintained at 9.6. Then 12 mL of TEOS was added dropwise into the solution. Next, metallic precursors calcium nitrate and zinc nitrate were added, respectively. The solution was kept under continuous stirring the whole time. Afterwards, the solution was left at a dry place for 4 hours to allow the reaction between reactants and then centrifuged at 8000 rpm for 10 min to collect the particles. The drying was done in an oven at 70 °C overnight. At the end, calcination of dried particles was performed in a muffle furnace at 700 °C for 5 hours.”
2- The description of volumes in line 124 and figure 1 is not in agreement with data in table 1. Furthermore, I suggest to indicate always the mL used and not vol% (scale-up results are unknown). Similarly, please write 6 g in line 123 (wt% as no meaning), and 37 mL in line 128.
Ans: Thank you for highlighting this. We regret the mistake on our behalf. The values given in line 124 and figure 1 are correct. We have corrected the values in the table 1 accordingly.
Table 1. Composition of zein/Zn-Mn MBGNs suspensions used to deposit composite coating.
|
Types of Suspensions |
Composition |
||||
|
Zn-Mn MBGNs (g/L) |
Zein (g/50mL) |
Acetic acid (mL/50 mL) |
Ethanol (mL/50 mL) |
Distilled water (mL/50mL) |
|
|
1 |
1 |
3 |
10 |
37 |
3 |
|
2 |
2 |
3 |
10 |
37 |
3 |
|
3 |
3 |
3 |
10 |
37 |
3 |
|
4 |
4 |
3 |
10 |
37 |
3 |
Similarly, the values mentioned as wt.% and vol.% are changed to the g and mL used, respectively (lines 134-136, 139, 146).
3- Please revise formula (1), line 181, A is in cm2 (deposition yield is in mg/cm2).
Ans: Thank you for the comment. The formula (1) is revised (line 192) as below.
4- In line 294, what do Authors mean with "flow properties"?
Ans: By flow properties we mean that the particles are in good spherical shape and they will move easily in the suspension under the influence of electric field. In literature, it is reported that the doping of metallic ions may vary the final shape of the particles (F. Westhauser et al), however, in this study the addition of metallic ions had no significant effect on the particles’ final shape.
Ref: F. Westhauser, S. Wilkesmann, Q. Nawaz, S. I. Schmitz, A. Moghaddam, and A. R. Boccaccini, “Osteogenic properties of manganese-doped mesoporous bioactive glass nanoparticles.,” J. Biomed. Mater. Res. A, vol. 108, no. 9, pp. 1806–1815, Sep. 2020, doi: 10.1002/jbm.a.36945
5- line 304 and following- (a) the positive charge value on the surface of ZnMn MBGNs should be reported; (b) The zeta-potential of the suspensions reported indicate that they have low colloid stability. Obviously, the non-stable particles will sediment rapidly. Please comment in the text how you are able to obtain suspensions; (c) why the zeta potential gets lower by adding positive Zn-Mn MBGNs?
Ans: As the reviewer suggested, we have added the zeta potential value for Zn-Mn MBGNs in line 341. We agree with the reviewer that the zeta potential of suspensions get lowered as compared to the pure zein and pure Zn-Mn MBGNs suspensions which always happens when particle loading is increased. However, lowering of zeta potential does not always lead to the rapid settling of particles (Rajesh et al). In our case, the suspensions were sufficiently stable during the whole time (5 min) of deposition and studies (Atiq et al and Nawaz et al) have shown the successful deposition even at lowered values of zeta potential. The explanation is also added in the manuscript (lines 341-347).
Ref: Rajesh Choudhary, Deepak Khurana, Aditya Kumar & Sudhakar Subudhi (2017) Stability analysis of Al2O3/water nanofluids, Journal of Experimental Nanoscience, 12:1, 140-151, DOI: 10.1080/17458080.2017.1285445
- Atiq Ur Rehman, Fatih Erdem Bastan, Bilal Haider, Aldo R. Boccaccini, Electrophoretic deposition of PEEK/bioactive glass composite coatings for orthopedic implants: A design of experiments (DoE) study, Materials & Design, Volume 130, 2017, Pages 223-230, ISSN 0264-1275, https://doi.org/10.1016/j.matdes.2017.05.045.
- Nawaz et al., “Ag-Mn doped mesoporous bioactive glass nanoparticles incorporated into the chitosan/gelatin coatings deposited on PEEK / bioactive glass layers for favorable osteogenic differentiation and antibacterial activity Materials Advances,” Mater. Adv., vol. 1, no. 5, pp. 1273–1284, 2020, doi: 10.1039/d0ma00325e.
6- line 405- thickness of 18 microns sounds very high for achieving good mechanical properties.
Ans: The comment is valid, however, the thickness of 18 µm is very much sufficient for achieving good mechanical properties. Values higher than this are reported in literature for polymeric coatings with good adhesion results (Atiq et al). Also, the polymer (zein) is a major constituent of the matrix and it acts as a binder which gives rise to the good adhesion properties to the coating.
- Atiq Ur Rehman, F. E. Bastan, B. Haider, and A. R. Boccaccini, “Electrophoretic deposition of PEEK/bioactive glass composite coatings for orthopedic implants: A design of experiments (DoE) study,” Mater. Des., vol. 130, pp. 223–230, 2017, doi: https://doi.org/10.1016/j.matdes.2017.05.045.
7- FTIR analysis is not convincing: (a) the bands at 800, 1040 and 1067 cm-1 attributed to Zn-MBGNs are present also in bare zein, figure 6; (b) lines 431-432 are not supported by data: please discuss why hydrogen bonding.
Ans: We thank the reviewer for the comment. The FTIR is performed again and new graph is added in the revised manuscript (figure 6). The peaks are distinguished for the two different materials. Further explanation of the FTIR is also added (line 520-532) in the revised manuscript. The added discussion is provided below.
“Pure zein represented characteristics peaks of amide I (C=O), II (N-H) and III (C-N) at 1650 cm–1, 1520 cm–1, and 1230 cm–1, respectively. After incorporating Zn-Mn MBGNs in zein, a similar pattern was observed in FTIR spectrum. The peaks attributed to the MBGNs were prominent at 800 cm–1 (Si-O-Si bending vibration), 1040 cm–1 (SiO2 stretching vibration), and 1067 cm–1 (Si-O-Si bending vibration). The aqueous medium of suspension causes Zn-Mn MBGNs to form free hydroxyl group on its surface which creates the possibility of H-bonding between the two materials [64][33]. The decreased intensity of C=O vibration after adding Zn-Mn MBGNs in zein indicates the H-bonding, due to which, the co-deposition of zein and Zn-Mn MBGNs was possible to attain. The strong H-bonding of the composite coating also resulted in excellent adhesion strength (discussed in next sections).”
8- lines 445-446 region 35-80 degrees is fully hydrophilic, please change the sentence.
Ans: The correction is made in the sentence (line 482). We meant to write that surface should not be super hydrophilic or super hydrophobic which may not be beneficial for protein attachment.
9- line 455, please change "augments the hydrophobic character of composite coating" into "alters the wettability of the surface".
Ans: Thank you for the comment. The correction is made in the sentence (line 492).
10- Paragraph 3.11 is poor: ionic release of Zn could support the discussion in line 525-526. Furthermore, the release of Mn and of the metals from the metallic support should be measured for better understanding biocompatibility of these materials.
Ans: We strongly agree with the comment. We have incorporated the study of ionic release from the MBGNs in the revised manuscript (section 2.6.7). The results (updated section 3.11) of Zn-ion release support the antibacterial effect which is further explained in the revised version. The biocompatibility study is also included in terms of effect of composite coating on MG-63 cell lines (sections 2.6.10 and 3.14).
11- lines 534-536- the discussion about the presence of P should be supported by a suitable reference.
Ans: As suggested by the reviewer, we have added further explanation and references to our argument related to the P (lines 647-653). The addition is also provided below.
“It was observed that HA started to form at day 3 (Figure 12A) as the composition of BG used in this study was composed of SiO2 and CaO in the ratio of 70:30 without any addition of phosphorous (P). The P is usually present in the form of Ca3(PO4)2. It regulates the formation and proliferation of osteobalsts while hindering the osteoclasts differentiation. Hence, P may be considered an accelerator in bioactivity and the delayed formation of HA could be due to absence of P in the BG [2]. Another possible reason for the delayed HA formation could be the release of Zn ions from the coating. Zn may get attached to the active growth sites of HA and retards the nucleation of HA in SBF [3]. However, HA was formed which proved the bioactivity of composite coating.”
12- In conclusions, point 4 is not supported by results.
Ans: We have made changes in the FTIR section of results and discussion which supports point 4 of the conclusions. The explanation can be seen in the answer of comment 7.

Reviewer 2 Report
Presented manuscript jfb-1805482 describes the Zn-Mn co-mesoporous bioactive glass nanoparticles loaded zein coatings on 316L stainless steel. The morphology, structure analysis, bioactivity and antibacterial effect of the coatings are presented and discussed. The optimization of the EPD parameters was done by the DoE array. The manuscript is well written, however, some minor issues appeared:
Line 128 page 3: 74 vol.% of absolute ethanol was added. It does not reflect figure 1 line 139. There is only 37 vol.% Ethanol is depicted in the figure.
Line 497 - Figure 8Error! Reference source not found.
Line 502 – Figure 8B: the y axis is not well visible.
Questions:
1) Authors explain the delay in HA formation in the absence of P. However, the Zn itself can lead to the postponed HA formation as well?
2) The bacterial inhibition is attributed to the presence of Zn in MBGNs. For this statement, it will be nice if the Zn ion release was measured. Even the changes in pH due to the dissolution products from the MBGNs can lead to bacterial inhibition and not Zn itself. The authors should even consider this possibility.
After this, the present manuscript can be accepted for publication.
Author Response
Reviewer 2
1- Line 128 page 3: 74 vol.% of absolute ethanol was added. It does not reflect figure 1 line 139. There is only 37 vol.% Ethanol is depicted in the figure.
Ans: Thank you for the comment. The values are corrected and highlighted in the revised manuscript (line 139).
2- Line 497 - Figure 8Error! Reference source not found.
Ans: The figure numbers are corrected in the revised manuscript.
3- Line 502 – Figure 8B: the y axis is not well visible.
Ans: The figure axis is re-adjusted for clear visibility of the readers. Please see figure 8 on page 18 of revised manuscript. The same is provided below.
Figure 8. Graphs showing the CoF Vs. sliding distance curves for (A) pure zein coating, (B) composite coating, (C) wear rate of pure zein and composite coatings. Wear track images taken from optical microscope (a) pure zein, (b) composite coating.
4- Authors explain the delay in HA formation in the absence of P. However, the Zn itself can lead to the postponed HA formation as well?
Ans: Thank you for the comment. We agree with the statement and this point is also added in the discussion part of the manuscript with suitable reference. Following explanation is incorporated to the manuscript (line 604-607).
“Another possible reason for the delayed HA formation could be the release of Zn ions from the coating. Zn may get attached to the active growth sites of HA and retards the nucleation of HA in SBF [51].”
5- The bacterial inhibition is attributed to the presence of Zn in MBGNs. For this statement, it will be nice if the Zn ion release was measured. Even the changes in pH due to the dissolution products from the MBGNs can lead to bacterial inhibition and not Zn itself. The authors should even consider this possibility.
Ans: We agree with the comment. The ionic release profile of Zn-Mn MBGNs is added (section 2.6.7 and 3.11) in the manuscript. The Zn-ion release is confirmed. The pH change could also lead to the bacterial inhibition. Following explanation is added in the manuscript (line 635-638), as per reviewers’ suggestion.
“The antibacterial effect also affirms the release of Zn-ion in the solution as indicated in the Zn-ion release profile (Figure 10). The dissolution products of MBGNs can also cause the death of bacteria via cell damage due to the increased pH [68].”

Reviewer 3 Report
This is a very interesting article about the deposition of pure zein and zein/Zn-Mn mesoporous bioactive glass nanoparticles (MBGNs) composite coatings on 316L stainless steel (SS) samples by means of electrophoretic deposition (EPD). However, some expository aspects of it and the analysis of the results need to be significantly improved (and corrected) to achieve the quality standards of this journal. Therefore, I recommend reconsidering it after major revisions.
1) Please improve the description of the electrochemical polarization tests. For example: how long were samples kept in open circuit potential conditions? What was the potential range swept during the potentiodynamic polarization test? How many samples were done for each analysis? In addition, please confirm how was the corrosion current density (icorr) obtained? By Tafel extrapolation method (please describe in detail the used procedure)? If yes, I would like to suggest that the authors take a look at the work of E, McCafferty (Corrosion Science 47 (2005) 3202–3215) and that they make some comments about the validity of their data (Mandatory!!!).
2) It is also recommended that the authors address the repeatability of their electrochemical results. What was the variation observed between the OCP and corrosion potential (Ecorr) measurements? What was the standard deviation of the icorr measurements? Also, for me it is not clear whether the data (presented in the inset of the Figure 9.) are single results or averages of multiple data. In these systems, it is well known that reproducibility of data is always an issue and the authors need to state this. (mandatory).
3) The analysis and discussion of polarization measurements are unfortunately very superficial and insufficient. It is mandatory to describe, at least, qualitatively the polarization behavior of the different samples. Also, please present the morphology of the corrosion attack after the polarization tests and make some comments about it.
4) In the manuscript, the authors say that “…It shows that Icorr value for composite coating has decreased ~100 turns. Therefore, the coating can improve the corrosion resistance of 316L SS …” I believe that this statement is not entirely true because, when analyzing the polarization curves, the zein/Zn-Mn MBGNs coated sample has a lower pitting potential than the bare sample. Could the authors explain this issue a little better??
5) As a final comment, I would like to suggest that the authors carry out long-term (i.e., long immersion time) electrochemical impedance spectroscopy (EIS) investigation of the electrochemical corrosion behavior of the zein/Zn-Mn MBGNs coatings. I say that because most reports published in the literature about the corrosion behavior of coating materials are mainly based on the corrosion potential, corrosion current density or even on the pitting potential, whose determination is commonly made through short-term (i.e., short immersion time) potentiodynamic anodic polarization tests, and the measured value represents the potential at which pit chemistry and dissolution rate of a metastable pit stabilize. A much better and more meaningful approach envisioning a biomedical application of the coatings would be to characterize the rate of damage accumulation under free corrosion conditions such as analyzing the pit distribution after a period of exposure or measuring the global surface response by EIS after a long period of free corrosion exposure.
Author Response
Reviewer 3
1- Please improve the description of the electrochemical polarization tests. For example: how long were samples kept in open circuit potential conditions? What was the potential range swept during the potentiodynamic polarization test? How many samples were done for each analysis? In addition, please confirm how was the corrosion current density (i) obtained? By Tafel extrapolation method (please describe in detail the used procedure)? If yes, I would like to suggest that the authors take a look at the work of E, McCafferty (Corrosion Science 47 (2005) 3202–3215) and that they make some comments about the validity of their data (Mandatory!!!)
Ans: We thank the reviewer for the comment. We have added details of corrosion studies in the revised manuscript (line 273-278) as following.
“Initially, an open circuit potential was determined for 30 min. All potentiodynamic polarization scans were recorded at 37 °C in simulated body fluid (SBF) electrolyte at 2.5 mV/s scan rate in the potential range of ±500 mV. The corrosion potential (Ecorr) and corrosion current density (Icorr) were measured directly by extrapolation of Tafel region. The tangents were drawn across the anodic and cathodic curves. At the intersection of tangents, lines were drawn along the x and y axes to obtain the values of Icorr and Ecorr, respectively. The corrosion study was done in triplicate for coated and uncoated samples and the average results are reported here.”
The referred work of E. McCafferty is very interesting and provides in-depth knowledge of validity of Tafel method in determining the Icorr and Ecorr of materials. According to our understanding, if there is well defined anodic and cathodic region occurring in the plot and the polarization curves are obtained in a steady-state then the Tafel extrapolation method is valid for corrosion analysis. This is true in our case, hence we consider that the results obtained in this study are valid.
2- It is also recommended that the authors address the repeatability of their electrochemical results. What was the variation observed between the OCP and corrosion potential (E corr) measurements? What was the standard deviation of the I corr measurements? Also, for me it is not clear whether the data (presented in the inset of the Figure 9.) are single results or averages of multiple data. In these systems, it is well known that reproducibility of data is always an issue and the authors need to state this. (mandatory).
Ans: We agree with the statement that the reproducibility of results is a problem in corrosion analysis. This point is stated in the revised manuscript. Further details of variation in OCP and Ecorr are also added. In our study, we performed multiple tests (in triplicate) on bare and coated samples. The trend of samples in all the tests was similar with a little variation, hence we reported the best results here. We also report here the revised paragraph (lines 559-577) related to the corrosion results and discussion and hope that this clears the corrosion behavior of our samples.
“The potentiodynamic polarization curves of bare 316L SS and zein/Zn-Mn MBGNs coated samples are given in Figure 9. It is a well-known fact that the corrosion test results are hard to reproduce due to the variation in testing environment. Even though the test is standardized, the testing arrangement (apparatus) and manual handling contributes to the collected data [69]. Hence, the results reported here are an average of the experiments performed thrice.
The OCP for bare SS varied rapidly from -0.22 V to -0.19 V and then remained al-most constant at -0.13 V up to 30 min and for composite coating, OCP varied from -0.25 V to -0.21 V before becoming stable at the latter value. After OCP stabilized, the polarization curves were obtained. The anodic polarization curve depicts the dissolution of iron and other alloying elements from 316L SS into respective ions. The cathodic branch of polarization curve shows the hydrogen evolution. The junction of anodic and cathodic branch shifted towards left in case of coated sample which indicates a decrease in Icorr value. Bare sample shows more abrupt significant increase in corrosion potential in passive region as compare to coated sample, which is attributed to the rapid dissolution of substrate. The measured Ecorr and Icorr values are given in the inset of the Figure 9. It shows that Icorr value for composite coating has decreased ~100 turns which is a good exhibition of enhanced corrosion resistance. Therefore, the composite coating can improve the corrosion resistance of 316L SS.”
3- The analysis and discussion of polarization measurements are unfortunately very superficial and insufficient. It is mandatory to describe, at least, qualitatively the polarization behavior of the different samples. Also, please present the morphology of the corrosion attack after the polarization tests and make some comments about it.
Ans: We regret that the polarization measurement results were not up to the mark in the first version of our manuscript. We have tried our best to interpret the results according to our understanding in revised manuscript. We are afraid that at this point it is not possible to include the morphology of corrosion attack on our samples as the tests were performed a long while ago. However, in our future studies on zein based system will surely include the morphological analysis of samples subjected to corrosion test.
4- In the manuscript, the authors say that “…It shows that Icorr value for composite coating has decreased ~100 turns. Therefore, the coating can improve the corrosion resistance of 316L SS …” I believe that this statement is not entirely true because, when analyzing the polarization curves, the zein/Zn-Mn MBGNs coated sample has a lower pitting potential than the bare sample. Could the authors explain this issue a little better??
Ans: The reviewer is absolutely right in this regard. It is true that the pitting corrosion occurs in the most polymeric coatings due to the presence of porosity. Porosity in coatings is required for cell attachment and growth. However, in biological system, our major concern is to lower the Icorr value which is clearly lowered in this study. Effect of change of potential is not valid in this case because in human body, Ecorr is not likely to change.
5- As a final comment, I would like to suggest that the authors carry out long-term (i.e., long immersion time) electrochemical impedance spectroscopy (EIS) investigation of the electrochemical corrosion behavior of the zein/Zn-Mn MBGNs coatings. I say that because most reports published in the literature about the corrosion behavior of coating materials are mainly based on the corrosion potential, corrosion current density or even on the pitting potential, whose determination is commonly made through shortterm (i.e., short immersion time) potentiodynamic anodic polarization tests, and the measured value represents the potential at which pit chemistry and dissolution rate of a metastable pit stabilize. A much better and more meaningful approach envisioning a biomedical application of the coatings would be to characterize the rate of damage accumulation under free corrosion conditions such as analyzing the pit distribution after a period of exposure or measuring the global surface response by EIS after a long period of free corrosion exposure.
Ans: We are grateful for the suggestion and we will perform a long-term electrochemical impedance spectroscopy (EIS) investigation on our samples. The corrosion behavior of biological coatings is indeed a significant aspect to analyze and control, hence we would like to take this opportunity and carry-out in-depth corrosive evaluation of zein based coatings in future.

Round 2
Reviewer 1 Report
The manuscript has been improved following suggestions. Thank you.
1- Authors forgot to modify figure 1: please revise written text in the figure according to revisions in the manuscript (vol%, wt% into mL, mg)
2- lines 367-369: please add comments in the text about stabilization of suspension due to electrostatic forces of attraction.
3- figure 10 (new): if the dots are connected, please provide a function, which allows description of the experimental results, otherwise provide approximation or just single points.
4- line 633: I think "Ca3(PO4)2" is a general formula indicating non-crystalline calcium orthophosphate. Since presumably it is not stoichiometric, please change the formula into "amorphous calcium orthophosphate"
Author Response
Comments:
- Authors forgot to modify figure 1: please revise written text in the figure according to revisions in the manuscript (vol%, wt% into mL, mg)
Figure 1 is modified according to the reviewer’s suggestion.
- lines 367-369: please add comments in the text about stabilization of suspension due to electrostatic forces of attraction.
We have added following lines (369-372) at the suggested place.
The FA between zein and Zn-Mn MBGNs causes the polymeric chain to envelop the MBGNs. The zein enveloped particles maintains repulsive forces among themselves in the suspension due to same cationic nature. Thus a state of stability is obtained in the suspension.
We hope that the mechanism behind suspension stability is now comprehensible.
- figure 10 (new): if the dots are connected, please provide a function, which allows description of the experimental results, otherwise provide approximation or just single points.
Figure 10 is modified.
- line 633: I think "Ca3(PO4)2" is a general formula indicating non-crystalline calcium orthophosphate. Since presumably it is not stoichiometric, please change the formula into "amorphous calcium orthophosphate"
"Ca3(PO4)2" is changed to the amorphous calcium orthophosphate.

Reviewer 3 Report
COMMENTS:
I checked all the authors' responses to the reviewer's comments and all the issues raised were adequately replied. Therefore, I recommend accepting the article.
Author Response
Many thanks for the positive feedback.